# Comparative Efficacy of Tyrosine Kinase Inhibitors and Antibody–Drug Conjugates in HER2-Positive Metastatic Breast Cancer Patients with Brain Metastases: A Systematic Review and Network Meta-Analysis

**DOI:** 10.3390/cancers14143372

**Published:** 2022-07-11

**Authors:** Yan Wang, Hangcheng Xu, Yiqun Han, Yun Wu, Jiayu Wang

**Affiliations:** Department of Medical Oncology, National Cancer Center/National Clinical Research Center for Cancer/Cancer Hospital, Chinese Academy of Medical Sciences and Peking Union Medical College, Beijing 100021, China; wangyan07425@163.com (Y.W.); xuhangcheng15@126.com (H.X.); hanyiqun803@163.com (Y.H.); wuyun0912@126.com (Y.W.)

**Keywords:** TKI, ADC, HER-2-positive, breast cancer brain metastasis, network meta-analysis

## Abstract

**Simple Summary:**

Comparisons between the efficacy of tyrosine kinase inhibitors (TKIs) and antibody–drug conjugates (ADCs) in treating HER2-positive breast cancer brain metastasis (BCBM) patients have not previously been conducted. We performed a systematic review and Bayesian-based network meta-analysis to pool the estimates of progression-free survival, overall survival, and incidence of central nervous system (CNS) disease progression. The current study indicated that both T-DXd and T-DM1 presented better efficacy than TKIs regarding survival outcomes. Treatments containing neratinib or T-DM1 tended to rank the best in reducing the recurrent rate of CNS. Our study provides more evidence for the clinical decision making for patients with HER2-positive BCBM. More high-quality studies with standardized entry criteria and comparable CNS-related endpoints are warranted in the future.

**Abstract:**

HER2-positive breast cancer brain metastasis (BCBM) is an important clinical problem. A systematic review and network meta-analysis were conducted to compare the efficacy of tyrosine kinase inhibitors (TKIs) and antibody–drug conjugates (ADCs), two categories of emerging agents in this field. We implemented a comprehensive literature search of PubMed, Embase, the Cochrane Library, ClinicalTrials.gov, and abstracts of oncology conferences. A network meta-analysis following Bayesian approaches was performed. Pooled hazard ratios (HRs) and odds ratios (ORs) with credible intervals (CrIs) were calculated to estimate progression-free survival (PFS), overall survival (OS), and the incidence of central nervous system (CNS) disease progression. Sixteen studies were included. Pairwise comparisons of PFS showed salient divergency between T-DXd and the physician’s choice of treatment (HR 0.17; 95% CrI 0.03–0.82) or afatinib (HR 0.14; 95% CrI 0.02–1.00). T-DXd and T-DM1 ranked first regarding PFS and OS, respectively, followed by TKI-containing regimens. The incidence of CNS disease progression was analyzed separately according to baseline BCBM status, among which neratinib-containing regimens were most likely to rank the best. In conclusion, ADCs including T-DXd and T-DM1 showed better efficacy than TKIs in the survival outcomes for HER2-positive BCBM patients. Treatments based on neratinib or T-DM1 revealed favorable results in reducing the recurrent rate of CNS.

## 1. Introduction

Breast cancer (BC) is a disease with high heterogeneity, and the prognosis is increasingly improving as novel treatments emerge [1]. Despite the favorable survival outcome for women with early-stage BC, the mortality rate is still high when distant metastasis occurs [2]. The brain is one of the most common metastatic sites of BC, and statistics indicate that BC brain metastasis (BCBM) accounts for a considerable portion of all malignant brain metastases [3]. Approximately 5.1% of the patients diagnosed with BC experience BCBM [4] and the proportion is continually rising due to enhanced screening techniques and prolonged overall survival (OS) [5]. BCBM poses a significant threat to both prognosis and quality of life (QoL) for patients [6].

Generally, BC is categorized into four molecular subtypes based on the status of the hormone receptor (HR) and human epidermal growth factor receptor 2 (HER2) [7]. For newly diagnosed patients, the incidence of BCBM is the highest among HR-negative HER2-positive and triple negative BC (TNBC) patients [8]. In fact, BCBM eventually develops in up to half of the patients with HER2-positive tumors [9], leading to a huge need for management. The current armamentarium against BCBM includes local therapy (such as surgery and radiotherapy), systemic therapy, and supportive care [5,10]. HER2-targeted therapy, as one of the subtype-specific treatments, is an increasingly important part of the systemic therapy for HER2-positive BC patients with BCBM. However, the existence of the blood–brain barrier (BBB) makes the brain a “sanctuary site” for cancer cells through its protective efflux systems [11], leading in turn to the limited efficacy of traditional anti-HER2 agents such as trastuzumab on BCBM due to their poor penetrability through the BBB [12].

Recently emerging agents including tyrosine kinase inhibitors (TKIs) and antibody–drug conjugates (ADCs) show promising prospects. TKIs block the downstream signaling pathways of the pan-HER family by binding to tyrosine kinases, thus suppressing the proliferation and metastasis of tumor cells. Moreover, the small-molecular property allows TKIs to penetrate through the BBB more efficiently, making it possible to prevent and treat BCBM [13]. Numerous randomized controlled trials (RCTs) investigating TKIs such as lapatinib, afatinib, neratinib, pyrotinib, and tucatinib are conducted in HER2-positive BC patients [14], and suggest favorable efficacy and safety. ADCs, which are designed to contain monoclonal antibodies (mAbs), a linker, and cytotoxic drugs, display both impressive antitumor activity and reduced systemic side effects [15]. Novel HER2-targeted ADCs including trastuzumab emtansine (T-DM1) and trastuzumab deruxtecan (T-DXd, also known as DS-8201) are implemented in the treatment of HER2-positive breast cancer, such as in the EMILIA and DESTINY-Breast03 studies [16,17], and explored in treating BCBM [18,19].

Although both TKIs and ADCs are making advances in the treatment of patients with HER2-positive BCBM, how to choose the best options for this population is still a subject of debate since no head-to-head comparisons exist so far. To update our current state of knowledge, we conducted this systematic review and network meta-analysis (NMA) to compare the efficacy between TKIs and ADCs.

## 2. Methods

This systematic review and NMA were conducted following the Preferred Reporting Items for Systematic Reviews and Meta-Analyses (PRISMA) and the extension statement for NMA (PRISMA-NMA) [20]. The study protocol was registered on PROSPERO (registration number: CRD42022323581).

### 2.1. Search Strategy and Selection Criteria

A comprehensive literature search was performed without language restrictions in PubMed, Embase, the Cochrane Library, and ClinicalTrials.gov before 5 May 2022. Annual conferences for organizations such as the American Society of Clinical Oncology (ASCO), the European Society of Medical Oncology (ESMO), the San Antonio Breast Cancer Symposium (SABCS) and the Chinese Society of Clinical Oncology (CSCO) were manually browsed from March 2019 to May 2022. The searched keywords included Medical Subject Headings (MeSH) such as “breast cancer”, “metastasis”, “HER2-positive”, “TKI”, “ADC”, and their corresponding free terms (including different agents), with the terms written in every possible combination. The detailed search strategy is elaborated in Appendix A.

The selection criteria were predefined as RCTs containing the regimens of TKIs or ADCs in the treatment of HER2-positive metastatic BC with available data regarding survival or recurrent outcomes in the central nervous system (CNS). The exclusion criteria were as follows: (1) nonrandomized, single-arm or retrospective trials; (2) negative or ambiguous HER2 status of enrolled patients; (3) no predefined targeted outcomes; (4) a subgroup analysis of RCTs (except for the BCBM subgroup); and (5) real-world experiences. Titles and abstracts were screened for study selection, and for possible eligible studies, the full texts were further accessed. Two investigators (Wang and Xu) proceeded with the above procedures independently.

### 2.2. Data Extraction and Quality Assessment

According to the prespecified protocol, two independent researchers (Wang and Xu) independently extracted and summarized the information from the included studies. The following items were collected: basic features (study names, first authors, time of publication, interventional or control measures, number of participants, and baseline BCBM status), and the outcomes of interest [progression-free survival (PFS), OS, and the recurrent rate of CNS disease]. Of particular note was that the baseline with no restricted BCBM status was divided into “no BCBM” and “non- or stable BCBM” according to the study design. The definitions of endpoints were listed as follows: (1) PFS is the time from randomization to the first date of recurrence or progression, or death from any cause; (2) OS is the time from randomization to death due to any cause; (3) recurrent rate of CNS disease refers to the proportion of patients with no or stable brain lesions at baseline but who suffered from CNS relapse (including the progression of existing lesions or newly emerged lesions).

The quality assessment of the included RCTs was performed utilizing the revised Cochrane risk-of-bias tool (RoB 2) with Review Manager (version 5.3; The Nordic Cochrane Centre, Copenhagen, Denmark). The bias domains included bias arising from the randomization process, bias due to deviations from the intended interventions, bias due to missing outcome data, bias in the measurement of the outcome, and bias in the selection of the reported result [21]. Each trial was separately scored as having high, low, or unclear risk by the two coauthors. The disagreement was solved by discussion.

### 2.3. Data Synthesis and Statistical Analysis

This study was based on the Bayesian framework that was carried out using the “gemtc” and “rjags” package of R software (version 4.1.1; R Foundation, Vienna, Austria). Two-tailed *p* < 0.05 was considered statistically significant. The magnitude of heterogeneity across the studies was quantified by the I^2^ index and Cochrane’s Q test. I^2^ > 50% and *p* < 0.10 for the Q test indicated substantial heterogeneity. The options for the fixed- or random-effects model employed to pool the effect sizes were determined by the heterogeneity. The hazard ratios (HRs) with 95% confidence intervals (CIs) and frequencies collected from the original trials were used to generate the combined relative efficacy for the survival data (PFS and OS), and categorial data (recurrent rate of CNS disease), respectively. The pooled-effect estimates were presented as HR or odds ratios (OR) with a corresponding 95% credible interval (CrI). The Markov chain Monte Carlo (MCMC) model was used to implement the above calculation. To be more precise, three Markov chains were used for simulation, and the number of tuning and simulating iterations was 10,000 and 50,000, respectively. The convergence of the model was estimated by the trace plots, density plots, and the Brooks–Gelman–Rubin diagnosis plot. Network plots were drawn to display the indirect and direct comparisons. The league tables and forest plots were generated to visualize the pairwise comparisons. All of the interventions were ranked by the surface under the cumulative ranking curve (SUCRA). The SUCRA values ranged between 0 and 100%, and also indicate the ranking of treatments from the worst to the best.

## 3. Results

### 3.1. Characteristics of the Included Study

A total of 4663 records were identified through the electronic databases; the detailed literature-screening process for these records is shown in Figure 1. A total of 20 articles involving 16 RCTs were eventually included in the NMA. The quality assessment of the included studies is shown in Figure 2. For survival and the recurrent outcomes of CNS, the basic information (such as the author, published time, study phase and intervention regimens) and predetermined endpoints were documented in Table 1 and Table 2, respectively. As for PFS or OS, there were 523 and 339 patients in the interventional and control group, respectively, while for the CNS progression rate, the enrollment numbers were 2640 and 2466, respectively.

The treatment strategies were summarized into distinctive broad categories, including chemotherapy (CT), physician’s choices (PC), mAb ± CT, TKI ± CT, and ADC ± CT. Thereinto, CT included capecitabine, taxane, or vinorelbine, and PC indicated any medical treatment approved for advanced or metastatic breast cancer. In this way, 13 different interventions were incorporated and compared in this study. The concrete agents were listed as follows: (1) mAb: trastuzumab; (2) TKIs: lapatinib, afatinib, neratinib, pyrotinib, and tucatinib; and (3) ADCs: T-DM1 and T-DXd.

### 3.2. Survival Outcomes

PFS and OS were two endpoints of interest regarding survival in this research. A total of nine and six interventions involving nine and five direct comparisons formed the networks of PFS and OS, respectively, as illustrated in Figure 3a,b. The pairwise comparisons between every two interventions are listed in Table 3. BCBM patients treated with T-DXd displayed significantly better PFS compared with PC (HR, 0.17; 95% CrI, 0.03–0.82). Additionally, T-DXd showed a clear trend in favoring PFS compared with afatinib monotherapy (HR, 0.14; 95% CrI, 0.02–1.00), although it did not reach significance. All other pairwise comparisons showed no salient difference.

The SUCRA values revealed the ranking of the efficacy of each treatment regimen (Table 4a,b). Regarding PFS, T-DXd was the most likely to rank the best with a SUCRA value of 91.45%, followed by TKI- or trastuzumab-containing therapies and T-DM1. Afatinib ± CT (16.31% or 23.56%) and PC (20.71%) seemed to rank the worst in terms of PFS. Among TKIs, neratinib (76.12%) and tucatinib (69.59%) were superior to lapatinib (57.33%) and afatinib (23.56%). As for OS, T-DM1 (86.14%) showed a conspicuous advantage over TKI-containing treatments including afatinib, neratinib, and lapatinib (32.51% to 46.32%). Furthermore, after combining different agents of the same kind, TKIs and ADCs had a similar level of efficacy for both PFS and OS on the whole as suggested in the forest plots (Appendix A). However, TKIs combined with CT revealed a tendency to perform worse than T-DM1 regarding OS (HR, 2.6; 95% CrI, 0.97–6.9) (Appendix A).

### 3.3. CNS Recurrent Rate

A total of nine interventions with 12 direct comparisons were incorporated into the NMA (Figure 3c). Considering the high heterogeneity in the baseline BCBM status of the included studies, we conducted the analysis in the overall population, the no BCBM subgroup, and the non- or stable BCBM subgroup, separately. Particularly, the available data in patients without any baseline brain lesions were limited, and the related data for patients with unstable BCBM were excluded. The SUCRA values for the recurrent rate of CNS are summarized in Table 4c.

The ranking order of the interventions was consistent across different groups on the whole, except for the subtle inconsistency of the sequence between afatinib plus CT and lapatinib plus CT in different situations. As suggested from the SUCRA values, neratinib with CT was the most likely to show the best control of CNS disease (80.15% for all patients and 77.70% for patients with non- or stable BCBM). T-DM1 combined with CT ranked higher than all the TKI-containing schemes (afatinib, lapatinib, and pyrotinib) other than neratinib, regardless of baseline BCBM status. Trastuzumab plus CT, T-DM1 and CT monotherapy presented obvious inferiority compared to other interventions (6.19–58.11%). The comparisons were further visualized by forest plots (Appendix A). Both neratinib and lapatinib with chemotherapy conspicuously diminished the recurrent or progressive rate of CNS lesions, no matter in the overall population (OR 0.19, 95% CrI 0.055–0.59; OR 0.31, 95% CrI 0.11–0.78) or in the non- or stable BCBM subgroup (OR 0.21, 95% CrI 0.04–0.88; OR 0.31, 95% CrI 0.095–0.90).

## 4. Discussion

HER2-positive BC is an aggressive subtype that was previously associated with poor prognosis. With the development of HER2-targeted pharmaceuticals since the last century, the outcomes of this entity have improved remarkably [40]. However, HER2-positive BC has a predisposition for developing BCBM [41], which remains a complex issue to tackle. An increasing number of clinical trials have focused on this topic, including the subgroup analyses of BCBM from pivotal RCTs [26,28] and specifically designed studies for BCBM [23]. TKIs and ADCs, as two categories of emerging agents, are evaluated in the BCBM settings without head-to-head comparisons between them. Therefore, we conducted this systematic review and NMA to figure out the optimal treatment regimens for patients with BCBM. The TKIs compared in this analysis included lapatinib, afatinib, neratinib, pyrotinib, and tucatinib, while the ADCs included T-DM1 and T-DXd. We explored the efficacy of the above agents on the PFS, OS, and the incidence of CNS disease progression.

Regarding survival outcomes, the pairwise comparisons did not show many significant differences except for the discrepancy between T-DXd and the physician’s choice of treatment in PFS. The SUCRA values that indicated the efficacy rank of different interventions further confirmed the salient advantage of T-DXd monotherapy over other treatment strategies, though the OS data on T-DXd were lacking. The higher drug-to-antibody ratio (DAR), stability in plasma, and short half-life payloads contribute to the potent antitumor activity of T-DXd [42]. At the same time, the increased permeability of the BBB and the blood–tumor barrier (BTB) after surgery, radiotherapy, and tumor invasion allows for the delivery of macromolecular agents to CNS lesions [43,44]. Moreover, the HER2 status is dynamic during the course of disease progression with the HER2 discordance between primary lesions and BCBM over 10% [45,46,47]. Apart from HER2-positive BC, T-DXd shows promising prospects in HER2-low BC [48], which could also partially explain the efficacy of T-DXd. By contrast, no evidence supports the use of TKIs or T-DM1 for treating HER2-low BC. In addition, there was also an obvious tendency for T-DXd to manifest better PFS than the afatinib monotherapy. As an irreversible blocker of the ErbB family, afatinib was proved effective in the treatment of EGFR-mutated non-small-cell lung cancer (NSCLC) brain metastases [49]. However, similar expected outcomes did not present in BCBM, which might be the result of lower afatinib concentration in the cerebrospinal fluid (CSF) [23]. In line with the study, the current pooled analysis is further proof that afatinib is not indicated for BCBM. T-DM1 was another ADC analyzed in this NMA. The phase III EMILIA trial laid the foundation for the use of T-DM1 in the second-line treatment of patients with HER2-positive advanced BC [17]. The exploration of T-DM1 in the BCBM setting also proceeded with satisfying results [31,50]. Here, we showed the comparatively superior efficacy of T-DM1 in both PFS and OS as indicated from the SUCRA values, especially in OS. Compared with trastuzumab, despite their similar macromolecular property and drug tissue distribution, T-DM1 led to the delayed growth of HER2-positive brain tumors and increased tumor-cell apoptosis in mouse models [51], which could explain the efficacy of T-DM1 in BCBM to some extent.

Within various TKIs, neratinib- and tucatinib-containing regimens revealed relatively favorable efficacy in PFS, although they were inferior to T-DXd. Neratinib and tucatinib are both second-generation oral TKIs, except the former is an irreversible pan-HER inhibitor while the latter is a selective HER2 inhibitor [52]. The phase III NALA trial and HER2CLIMB trial both allowed for the enrollment of HER2-positive metastatic BC patients with BCBM [25,26,27]. Particularly, the HER2CLIMB trial also included patients with active (progressive and/or untreated) BCBM [27]. In view of the gratifying outcomes, neratinib plus capecitabine and the tucatinib–trastuzumab–capecitabine triplet are recommended for the later-line treatment for HER2-positive BC in metastatic setting, especially for patients with CNS disease [53]. In this NMA, we focused on the BCBM subgroup of the original studies. The collected data indicated that the neratinib combination did not show a significant benefit in PFS and OS, though it demonstrated significant improvement regarding PFS in the whole population [25]. In contrast, the tucatinib-containing regimen displayed both PFS and OS improvement in the BCBM subgroup compared with the control group. The remarkable efficacy coupled with the specific design for BCBM make tucatinib a preferentially recommended TKI for patients with BCBM compared to neratinib [53]. However, based on the current analysis, neratinib seemed to present better in PFS than tucatinib (SUCRA values: 76.12% vs. 69.59%). However, it should also be noted that tucatinib was not included in the network of OS; thus, the comparison between neratinib and tucatinib regarding OS remains unknown. Moreover, consideration should also be given to the safety profiles when selecting a proper treatment strategy, since more off-target adverse events might be encountered in neratinib than tucatinib.

Apart from survival outcomes, the incidence of CNS disease progression was another endpoint of interest in this study. BCBM could cause various neurological symptoms due to the compression of the brain tissue, which is closely related to poor QoL. Given the relatively dismal prognosis of patients with BCBM, to delay the progression of CNS disease and then improve QoL is of particular significance. Of note, in line with the conclusions from a recent published review by Talvinder Bhogal et al. [54], there was a high level of heterogeneity in the entry criteria and baseline BCBM status of the enrolled patients, which would confound the comparison. To reduce the potential bias, the included studies were divided according to the baseline BCBM status and the analysis of CNS disease recurrence, respectively. The synthesized results revealed that there was a consistent overall trend both in the whole population and in the subgroup population. To be specific, neratinib-containing regimens were most likely to have the best SUCRA values, followed by T-DM1 or TKIs plus chemotherapy, while monotherapies were the worst. In the NALA trial, health-related QoL, as one of the secondary endpoints, was reported to be generally maintained and symptoms such as headaches were less common in the patients treated with neratinib (10.6%) than lapatinib (16.4%) [25]. These results could reflect the close relationship between the QoL and CNS progression rate from the side. In addition, our present analysis further confirmed the advantages of neratinib over other interventions. Moreover, another conclusion that could be derived is that targeted therapy combined with chemotherapy showed the best efficacy among systemic therapies in postponing or potentially preventing BCBM.

Inevitably, several limitations exist in this systematic review and NMA. Firstly, not all the included studies consisted of all the outcomes of interest, leading to a lack of comparisons regarding a particular intervention in some situations. Thus, it is difficult to draw a big-picture view of the issue. Secondly, as mentioned above, the enrolled RCTs were divergent with each other in terms of the inclusion criteria and reported endpoints [54], which could explain the wide CrIs in the forest plots to some extent. However, limited by the small number of RCTs focusing on BCBM patients and the diverse trial designs, the above limitations were difficult to overcome. Thirdly, the SUCRA values could merely reflect the intervention rankings, lacking the ability to present the degree of absolute difference. Caution is warranted when applying the research findings to clinical practice since the treatment options are determined by multiple factors. In spite of the deficiencies, the general low I2 values indicated the acceptable heterogeneity and satisfying credibility of our analysis.

Altogether, this study combined evidence from current RCTs of high quality, which might be beneficial to future clinical practice with respect to the HER2-directed therapy for HER2-positive BC patients with BCBM. Recommendations could be made to the patients based on present efficacy outcomes, safety profiles, financial circumstances, and drug accessibility. Considering that the data of the extant studies continue to mature and many new trials in this field are ongoing, the future clinical decisions will certainly be constantly updated. Particularly, we have faith in T-DXd in treating BCBM due to its potent efficacy and manageable toxicity. The current 3b/4 DESTINY-Breast12 (NCT04739761) study that is in its recruiting phase will shed more light on the utilization of T-DXd in BCBM. At the same time, more attention should be paid to the distinct genomic landscape of BCBM [45] as well as the predictive value of serum microRNAs [55] and tumor-infiltrating lymphocytes (TILs) [56] for treatment and prognosis, which could make it possible to develop novel agents and identify potential biomarkers. Without a doubt, the standardization of inclusion criteria and CNS-related endpoints of future clinical trials is essential to the comparison and data synthesis of different treatment strategies for HER2-positive patients with BCBM.

## 5. Conclusions

In conclusion, ADCs tended to show better efficacy than TKIs regarding survival outcomes in HER2-positve BC patients with brain metastases. In this study, T-DXd displayed conspicuous superiority in PFS based on current data, while potential OS benefits were detected in T-DM1. Regimens containing neratinib and T-DM1 showed favorable rankings in postponing or potentially preventing BCBM.

## Figures and Tables

**Figure 1 cancers-14-03372-f001:**
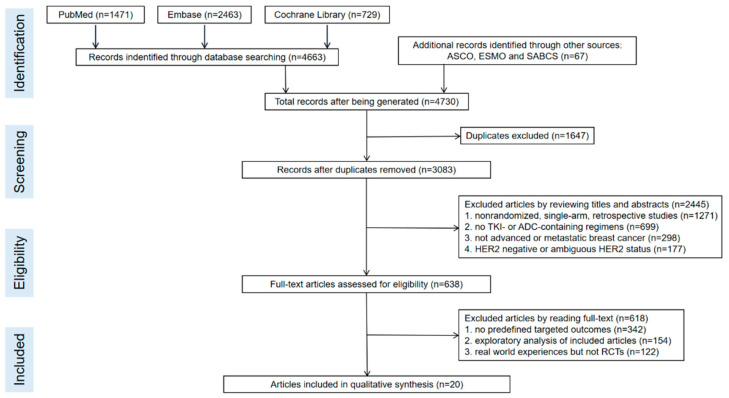
The flow chart of the detailed literature-screening process.

**Figure 2 cancers-14-03372-f002:**
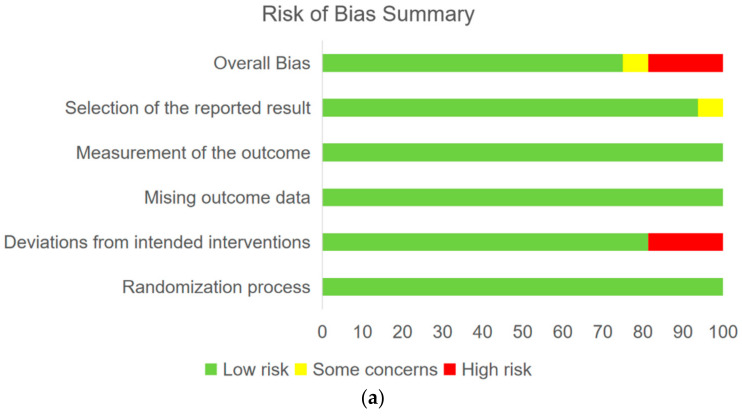
Quality assessment for the bias items of RCTs. (**a**) Risk of the bias summary. (**b**) Risk of the bias graph.

**Figure 3 cancers-14-03372-f003:**
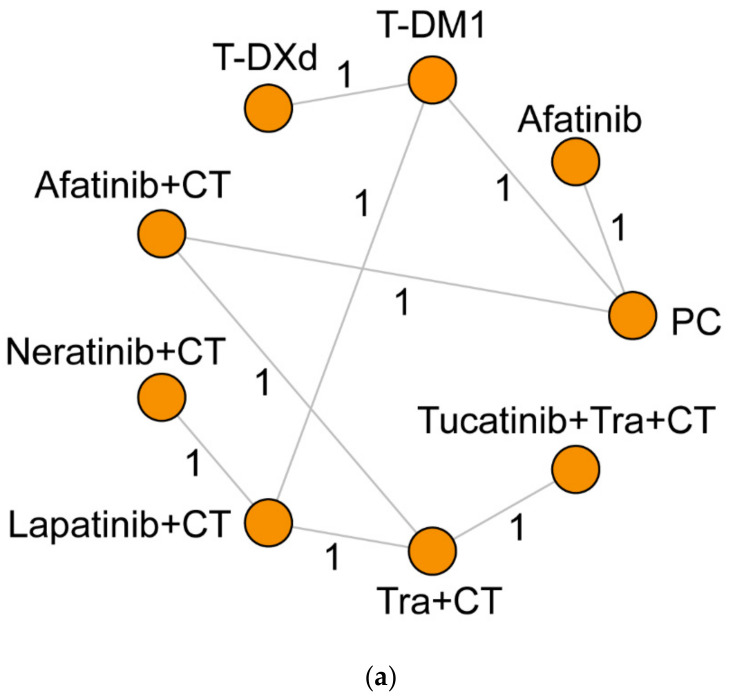
Network plots of PFS (**a**), OS (**b**) and the recurrent rate of CNS (**c**). Every node represents one kind of intervention while the lines are related to direct comparisons. Abbreviations: CNS, central nervous system; T-DM1, trastuzumab emtansine; T-DXd, trastuzumab deruxtecan; PC, physician’s choice; CT, chemotherapy; Tra, trastuzumab.

**Table 1 cancers-14-03372-t001:** The baseline characteristics and survival outcomes of the included RCTs.

Study	Author. Published Time	Phase	Regimen (No. of Patients)		Median PFS for BCBM (HR, 95% CI)	Median OS for BCBM (HR, 95% CI)
			Interventional Group	Control Group		
WJOG6110B/ELTOP [22]	Toshimi Takano et al., 2018.08	II	Lapatinib + Capecitabine (6)	Trastuzumab + Capecitabine (7)	NA vs. NA 0.62 (0.18–2.17)	-
LUX-Breast 3 (a) * [23]	Javier Cortés et al., 2015.12	II	Afatinib alone (40)	Physician’s Choice ** (43)	11.9 w vs. 18.4 w 1.18 (0.72–1.93)	57.7 w vs. 52.1 w 1.27 (0.72–2.21)
LUX-Breast 3 (b) * [23]	Javier Cortés et al., 2015.12	II	Afatinib + Vinorelbine (38)	Physician’s Choice ** (43)	12.3 w vs. 18.4 w 0.94 (0.57–1.54)	37.3 w vs. 52.1 w 1.60 (0.93–2.76)
LUX-Breast 1 [24]	Nadia Harbeck et al., 2016.03	III	Afatinib + Vinorelbine (43)	Trastuzumab + Vinorelbine (17)	NA vs. NA 1.32 (0.67–2.59)	-
NALA [25,26]	Cristina Saura et al., 2020.09/2021.08	III	Neratinib + Capecitabine (51)	Lapatinib + Capecitabine (50)	7.8 m vs. 5.5 m 0.66 (0.41–1.05)	16.4 m vs. 15.4 m 0.90 (0.59–1.38)
HER2CLIMB [27,28,29]	Nancy U. Lin et al., 2020.02/2020.08/2022.03	III	Tucatinib + Trastuzumab + Capecitabine (198)	Trastuzumab + Capecitabine (93)	7.6 m vs. 5.4 m 0.48 (0.34–0.69)	NA vs. NA 0.60 (0.44–0.81)
TH3RESA [30]	Ian E Krop et al., 2014.06/2017.06	III	T-DM1 (40)	Physician’s Choice ** (27)	5.8 m vs. 2.9 m 0.47 (0.2–0.89)	17.3 m vs. 12.6 m 0.62 (0.34–1.13)
EMILIA [17,31]	Sunil Verma et al., 2012.12/2015.01	III	T-DM1 (45)	Lapatinib + Capecitabine (50)	5.9 m vs. 5.7 m 1 (0.54–1.84)	26.8 m vs. 12.9 m 0.38 (0.18–0.80)
DESTINY-Breast03 [16]	Javier Cortés et al., 2022.03	III	T-Dxd (62)	T-DM1 (52)	15.0 m vs. 5.7 m 0.38 (0.23–0.64)	-

* LUX-Breast 3 was divided into two cohorts because the comparisons were between two kinds of afatinib-containing regimens and the physician’s choice of therapy. ** Physician’s choice included any systemic therapy (including chemotherapy, hormonal therapy, and HER2-directed therapy) approved for advanced or metastatic breast cancer. Abbreviations: NA, not available; PFS, progression-free survival; OS, overall survival; HR, hazard ratio; CI, confidence interval; m, months; w, weeks.

**Table 2 cancers-14-03372-t002:** The baseline characteristics and the recurrent outcomes of CNS of the included RCTs.

Study	Author. Published Time	Phase	Regimen (No. of Patients)		Incidence of CNS Disease Progression		Baseline Characteristic
			Interventional Group	Control Group	Interventional Group	Control Group	
NCT00078572 [32]	Charles E. Geyer et al., 2006.12	III	Lapatinib + Capecitabine (163)	Capecitabine (161)	4/163	11/161	Non- or stable BCBM *
EGF100151 [33]	David Cameron et al., 2008.12	III	Lapatinib + Capecitabine (198)	Capecitabine (201)	4/198	13/201	Non- or stableBCBM *
CEREBEL [34]	Xavier Pivot et al., 2015.05	III	Lapatinib + Capecitabine (251)	Trastuzumab + Capecitabine (250)	17/251	15/250	No BCBM
WJOG6110B/ELTOP [22]	Toshimi Takano et al., 2018.08	II	Lapatinib + Capecitabine (43)	Trastuzumab + Capecitabine (43)	2/43	2/43	Non- or stable BCBM *
NCIC CTG MA.31 [35]	Karen A. Gelmon et al., 2015.05	III	Lapatinib + Taxane (242)	Trastuzumab + Taxane (219)	40/270	48/267	No BCBM
LUX-Breast 1 [24]	Nadia Harbeck et al., 2016.03	III	Afatinib + Vinorelbine (339)	Trastuzumab + Vinorelbine (169)	30/339	19/169	Non- or stable BCBM *
NCT00777101 [36]	Miguel Martin et al., 2013.12	II	Neratinib (117)	Lapatinib + Capecitabine (116)	11/117	15/116	Non- or stable BCBM *
NEfERT-T [37]	Ahmad Awada et al., 2016.12	II	Neratinib + Paclitaxel (242)	Trastuzumab + Paclitaxel (237)	20/242	41/237	Non- or stable BCBM *
NALA [25]	Cristina Saura et al., 2020.09/2021.08	III	Neratinib + Capecitabine (307)	Lapatinib + Capecitabine (314)	70/307	92/314	Non- or stable BCBM *
PHOEBE [38]	Binghe Xu et al., 2021.03	III	Pyrotinib + Capecitabine (134)	Lapatinib + Capecitabine (132)	3/134	3/132	No BCBM
EMILIA [17,31]	Sunil Verma et al., 2012.12/2015.01	III	T-DM1 (495)	Lapatinib + Capecitabine (496)	19/495	11/496	Non- or stable BCBM *
NCT01702558 [39]	Javier Cortés et al., 2020.08	I/II	T-DM1 + Capecitabine (81)	T-DM1 (80)	3/81	7/80	Non- or stable BCBM *

* The baseline “non- or stable BCBM” represented “non- or stable (treated or untreated), asymptomatic CNS lesions”.

**Table 3 cancers-14-03372-t003:** Pairwise comparisons among nine interventions for PFS and OS (HR, 95% CrI).

PC	1.17 (0.39, 3.57)	0.43 (0.14, 1.33)	**0.17** **(0.03, 0.82)**	0.93 (0.33, 2.63)	0.28 (0.05, 1.66)	0.43 (0.11, 1.71)	0.70 (0.19, 2.69)	0.34 (0.06, 1.89)
0.80 (0.21, 2.96)	Afatinib	0.37 (0.08, 1.80)	0.14 (0.02, 1.00)	0.80 (0.17, 3.62)	0.24 (0.03, 1.94)	0.37 (0.07, 2.08)	0.60 (0.11, 3.43)	0.29 (0.04, 2.25)
1.60 (0.43, 6.09)	2.03 (0.32, 13.41)	T-DM1	0.39 (0.12, 1.20)	2.18 (0.57, 8.32)	0.65 (0.14, 3.10)	1.01 (0.35, 2.95)	1.64 (0.39, 6.79)	0.79 (0.13, 4.76)
-	-	-	T-DXd	5.68 (0.93, 31.89)	1.71 (0.24, 11.92)	2.61 (0.54, 12.95)	4.25 (0.68, 26.56)	2.05 (0.24, 17.40)
0.62 (0.17, 2.33)	0.79 (0.12, 5.04)	0.38 (0.06, 2.41)	-	Afatinib + CT	0.30 (0.05, 1.86)	0.46 (0.12, 1.90)	0.76 (0.25, 2.20)	0.37 (0.08, 1.62)
0.68 (0.07, 6.89)	0.85 (0.06, 12.35)	0.42 (0.06, 2.76)	-	1.08 (0.08, 15.65)	Neratinib + CT	1.54 (0.49, 4.75)	2.49 (0.44, 13.56)	1.21 (0.16, 9.05)
0.61 (0.09, 4.01)	0.77 (0.08, 7.64)	0.38 (0.09, 1.48)	-	0.98 (0.10, 10.16)	0.91 (0.25, 3.12)	Lapatinib + CT	1.63 (0.45, 5.81)	0.79 (0.15, 4.12)
-	-	-	-	-	-	-	Tra + CT	0.49 (0.16, 1.46)
-	-	-	-	-	-	-	-	Tucatinib + Tra + CT

Remarks: The summary of PFS pairwise comparisons (on the upper triangle with light blue backgrounds) and OS pairwise comparisons (on the lower triangle with white backgrounds) from the network meta-analysis. Every square has the corresponding horizontal and vertical interventions (with light brown backgrounds), in which every figure reveals the preferential tendency for the longitudinal agents if lower than 1. A statistically significant difference was bolded in red. Abbreviations: CI, confidential interval; HR, hazard ratio; PFS, progression-free survival; OS, overall survival; PC, physician’s choice; CT, chemotherapy; Tra, trastuzumab.

**Table 4 cancers-14-03372-t004:** SUCRA values for different interventions of PFS (**a**), OS (**b**) and the recurrent rate of CNS (**c**).

(a) PFS			
Interventions			SUCRA Values (%)
T-DXd			91.45
Neratinib + CT			76.12
Tucatinib + Tra + CT			69.59
T-DM1			58.89
Lapatinib + CT			57.33
Tra + CT			36.04
Afatinib + CT			23.56
PC			20.71
Afatinib			16.31
**(b) OS**			
**Interventions**			**SUCRA Values (%)**
T-DM1			86.14
PC			61.50
Afatinib			46.32
Neratinib + CT			40.49
Afatinib + CT			33.04
Lapatinib + CT			32.51
**(c) Recurrent rate of CNS**			
**Interventions**	**SUCRA Values (%)**		
	**All patients**	**No Baseline BCBM**	**Baseline Non- or** **Stable BCBM**
Neratinib + CT	80.15	-	77.70
Neratinib	71.00	-	72.61
T-DM1 + CT	68.50	-	69.45
Afatinib + CT	56.49	-	50.57
Lapatinib + CT	52.79	67.77	56.16
Pyrotinib + CT	52.72	62.03	-
Tra + CT	38.75	58.11	35.51
T-DM1	23.41	12.08	28.60
CT	6.19	-	9.41

Abbreviations: SUCRA, the surface under the cumulative ranking curve; PFS, progression-free survival; OS, overall survival; CNS, central nervous system; BCBM, breast cancer brain metastasis; T-DXd, trastuzumab deruxtecan; T-DM1, trastuzumab emtansine; CT, chemotherapy; Tra, trastuzumab.

## Data Availability

The data presented in this study are available on request from the corresponding author.

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
