# Peer review of "Comparative Efficacy of Tyrosine Kinase Inhibitors and Antibody–Drug Conjugates in HER2-Positive Metastatic Breast Cancer Patients with Brain Metastases: A Systematic Review and Network Meta-Analysis"

_cancers, 2022, doi:10.3390/cancers14143372_

Round 1

Reviewer 1 Report

The authors have responded appropriately to the reviewer's comments, and I consider them acceptable without further comments.

Author Response

We sincerely thank for your excellent work and recognition for our research.

Reviewer 2 Report

The authors addressed all the issues raised by the reviewers.

Author Response

We deeply appreciate your meticulous work and consideration of our manuscript.

Reviewer 3 Report

I thank the authors for their sincere response. I understand authors’ response to my comment, and I consider this paper could be worthy of publication. Before it, however, I would like to request a detailed explanation of one of the conclusions in the paper.

In conclusion, it says ‘Treatments based on neratinib or T-DM1 revealed favorable results in reducing the recurrent rate of CNS.’. However, given the SUCRA value of 23.41 for T-DM1 in Table 4(C), I do not believe T-DM1 is highly effective in CNS prophylaxis. Could you be more specific about your considerations on this point?

Author Response

Thanks for your excellent work. We are very privileged to respond to the concerns about the sentence “Treatments based on neratinib or T-DM1 revealed favorable results in reducing the recurrent rate of CNS”. It is noteworthy that the SUCRA values of T-DM1 in CNS prophylaxis was relatively low (12.08% to 28.60% in different groups). In our previous manuscript, we only focused on the interventions with top 3 SUCRA values, including neratinib plus chemotherapy, neratinib and T-DM1 plus chemotherapy. We simply summarized them as “treatments based on neratinib or T-DM1”. However, there existed inaccuracy and we have changed the sentence to “Treatments based on neratinib revealed favorable results in reducing the recurrent rate of CNS” (please see the last sentence in Abstract), considering the conspicuous divergence between T-DM1 monotherapy and T-DM1 plus chemotherapy. Additionally, the superiority of neratinib over other interventions can be confirmed by many other studies. This finding was also emphasized and discussed in the discussion part (please see the fourth paragraph in Discussion). Thank you again for your effort. The revised parts were marked in red in the latest version of manuscript. We sincerely hope you could reconsider our manuscript.

This manuscript is a resubmission of an earlier submission. The following is a list of the peer review reports and author responses from that submission.

Round 1

Reviewer 1 Report

This paper reports about the comparison of efficacy between tyrosine kinase inhibitor and anti-body drug conjugate in HER2-positive metastatic breast cancer patients with brain metastasis through a systematic review and network meta-analysis. There are many treatment options for HER2-positive breast cancer, and I think this topic is very important. I could not find any faults in the process of systematic review and network meta-analysis. However, there were very few trials included in the current study. I think it is a critical matter in the study.  The samples size in each trial were too small though the background of patients had heterogeneity in each trial. The conclusion might be poorly supported by the data. Unfortunately, I decided to inform the editor that this paper should not be accepted as what it was.

Reviewer 2 Report

This analysis has been corrected from a methodological point of view.

I simply request some language amendments.

Reviewer 3 Report

1.      I could not find a simple summary, summary or keywords specific to this paper.

2.      As for the Quality evaluation chart in Figure 2, I think it would be more helpful to readers if you added the name of the study as well as the name of the author. I also recommend that the order of the names be consistent with the order of Table 1 and Table 2

3.      I confirmed that important clinical studies related to TKIs and ADCs have been picked up without over- or under-delivery. No specific comments on the results of the analysis or the content of the discussion.

4.      It is difficult to distinguish between the three different tables a, b, and c included in Table 4. I would like to recommend adding notation such as PFS, OS, etc. in the table.